# The Gendered Experience of Close to Community Providers during COVID-19 Response in Fragile Settings: A Multi-Country Analysis

**Joanna Raven** [1,*], **Abriti Arjyal** [2], **Sushil Baral** [2], **Obindra Chand** [2], **Kate Hawkins** [3], **Lansana Kallon** [4], **Wesam Mansour** [1], **Ayuska Parajuli** [2], **Kyu Kyu Than** [5], **Haja Wurie** [4], **Rouham Yamout** [6] and **Sally Theobald** [1]

1   Department of Public Health, Liverpool School of Tropical Medicine, Liverpool L3 5QA, UK
2   HERD International, Kathmandu 44600, Nepal
3   Pamoja Communications, Brighton BN2 4AY, UK
4   College of Medicine and Allied Health Sciences, University of Sierra Leone, Freetown 00232, Sierra Leone
5   Burnet Institute, Yangon 3004, Myanmar
6   Faculty of Health Sciences, American University of Beirut, Beirut P.O. Box 11-0236, Lebanon
*   Correspondence: joanna.raven@lstmed.ac.uk

**Abstract:** Many countries, and particularly those including fragile contexts, have a shortage of formal health workers and are increasingly looking to close-to-community (CTC) providers to fill the gap. The experiences of CTC providers are shaped by context-embedded gender roles and relations. This qualitative research study in Lebanon, Nepal, Myanmar and Sierra Leone explored the gendered experiences of CTC providers during the COVID-19 pandemic in fragile settings. We used document review, in-depth interviews or focus group discussions with CTC providers, and key informant interviews with local stakeholders to generate in-depth and contextual information. The COVID-19-associated lockdowns and school closures brought additional stresses, with a gendered division of labour acutely felt by women CTC providers. Their work is poorly or not remunerated and is seen as risky. CTC providers are embedded within their communities with a strong willingness to serve. However, they experienced fractures in community trust and were sometimes viewed as a COVID-19 risk. During COVID-19, CTC providers experienced additional responsibilities on top of their routine work and family commitments, shaped by gender, and were not always receiving the support required. Understanding their experience through a gender lens is critical to developing equitable and inclusive approaches to support the COVID-19 response and future crises.

**Keywords:** close-to-community providers; gender; justice; fragile settings; COVID-19; community health workers; health policy

## 1. Introduction

The workforce is a key component of the health system that underpins Universal Health Coverage efforts (WHO 2016). Most countries in the global South, and particularly fragile settings, have a shortage of formal health workers and are increasingly looking to a range of close-to-community (CTC) providers (e.g., Community Health Workers and Traditional Birth Attendants) to fill the gap and reach the most marginalised (Miyake et al. 2017; Casey et al. 2015; Orya et al. 2017; Edmond et al. 2018; Raven et al. 2020). A CTC provider is a health worker who carries out promotional, preventive and/or curative health services and is the first point of contact at community level. A CTC provider can be based in the community or in a primary health care facility (REACHOUT n.d.).

The COVID-19 pandemic has demonstrated the value of CTC providers and their importance in responding, offering support and care (ARISE Hub 2020; Knox-Peebles 2020). A recent review identified the importance of role clarity, training, supportive supervision,

work satisfaction, health and well-being as crucial to the pandemic response as well as the need for more research focusing on gender and equity (Bhaumik et al. 2020).

COVID-19 has shone a spotlight on how gender shapes vulnerabilities and the pandemic response (Wenham et al. 2020). Gender roles and relations are context-embedded and dynamic, shaping the social determinants of health as well as the experiences of CTC providers (Steege et al. 2018; Raven et al. 2022). Understanding and addressing these roles and relations is important in appropriately supporting CTC providers as they navigate new and existing challenges. The ReBUILD for Resilience (ReBUILD) research programme focuses on health systems resilience in fragile settings—Lebanon, Myanmar, Nepal and Sierra Leone. As part of this programme, a research study on the gendered experience of CTC providers in these countries during the COVID-19 outbreak was commissioned.

A range of CTC providers were part of the COVID-19 response. In Lebanon, there is inadequate access to health services for both Lebanese and Syrian refugees. As a result, many refugees rely on outreach services and informal health facilities who employ health workers belonging to the Syrian community who are forbidden from practicing in Lebanon. They are informal health workers including CTC providers (Yamout and Khalil 2021). In Myanmar, there are 50,000 CTC providers including female auxiliary midwives and male community health workers delivering maternal and child health and disease control services. As part of the COVID-19 response, all CTC providers and community volunteers work in quarantine facilities, distribute food, and conduct hygiene activities (Ministry of Health and Sports Myanmar 2020). In Nepal, around 50,000 Female Community Health Volunteers (FCHVs) working as CTC providers support health promotion and prevention activities at community level. National guidelines called for FCHVs to be part of the COVID-19 response (Ministry of Health and Population Nepal 2019; Office of Prime Minister and Council of Ministers 2020). In Sierra Leone, there are 15,000 CTC providers across the country who provide a basic package of services mainly centred around maternal and child health, and they took on additional health education and preventive care roles during the COVID-19 pandemic (ReBUILD for Resilience Consortium 2021). At the time of data collection (December 2020), the four study countries were all experiencing the COVID-19 pandemic, with rapid transmission and increasing numbers of people with the infection and deaths (WHO 2022).

A recent review of global literature on CTC providers' gendered experiences during the COVID-19 pandemic in fragile settings found no associated empirical studies (Mansour and Raven 2021). Most papers were commentaries or opinion pieces, with some focusing on lessons learned from previous outbreaks and how to apply them during the COVID-19 pandemic, and patchy evidence on CTC providers' experiences. No differences between the roles and responsibilities of women and men CTC providers were reported in the literature. However, women were reported to be at an increased risk of exposure and infection, with limited access to Personal Protective Equipment (Ssali 2020; Nepomnyashchiy et al. 2020). Some studies found that women CTC providers put their families at greater risk of infection, because of family and household responsibilities, and reported women isolating themselves from their children (Muraya 2020). Public transport bans and curfews posed mobility challenges for CTC providers especially for those without cars or who worked in facilities with no vehicles, and this was especially the case for women (Ssali 2020; Devi 2020; Nanda et al. 2020). Women CTC providers faced additional risks of physical and verbal assault. Pairing women and men CTC providers was a strategy used in some areas to protect women (UNICEF, Save the Children, and International Rescue Committee 2020). Digital technology can enable real-time support for decision-making, supervision, community-based surveillance, contact tracing during COVID-19, but there was insufficient access to telephone and data networks for both women and men CTC providers (Nepomnyashchiy et al. 2020; UNICEF, Save the Children, and International Rescue Committee 2020). CTC providers often described feeling alone, disrespected, stigmatised and being viewed as 'carriers of the disease' (Bhaumik et al. 2020; Srinivasan and Arora 2020).

There is a critical gap in the local and global evidence base on CTC providers and COVID-19 and a need to develop this research area and strengthen community health systems (Nepomnyashchiy et al. 2020). This paper explores the roles of CTC providers and their gendered experiences during the COVID-19 pandemic and in the often-neglected contexts of fragile settings. This study contributes evidence on gender equitable approaches to supporting CTC providers to fulfil their vital role in the COVID-19 response and future crises and emergencies, which test fragile health structures and systems.

## 2. Materials and Methods

We used a qualitative research study design to generate in-depth and contextual information (Snape and Spencer 2003) on experiences of CTC providers during COVID-19 in Lebanon, Myanmar, Nepal and Sierra Leone. We used three methods: document review, in-depth interviews (IDIs) or focus group discussions (FGDs) with CTC providers, and key informant interviews (KIIs) with local stakeholders. We used gender analysis (Morgan et al. 2016) and gender and CTC provider frameworks (Steege et al. 2018) to guide our data collection, analysis and synthesis.

In each country, we purposively selected districts or communities based on accessibility and established relationships with local health stakeholders. This facilitated identifying participants and was a pragmatic approach to enable data collection within the challenges posed by the COVID-19 outbreak. In Lebanon we conducted the research in two sites in Beqaa Valley (peri-urban setting) where many Syrian refugees and vulnerable Lebanese people live. The first site is the Multi-Aid Programs (MAPS) organization which has three health centres and serves 13,000 people, most of whom are Syrian refugees and poor Lebanese people. The second site is a grouping of organizations implementing outreach activities in Barrelias, Al Marj, and Arsal areas of Beqaa Valley. In Myanmar, South Dagon township in Yangon Region (the largest region in Myanmar with a population of 5.3 million) was chosen because of the high number of people with COVID-19 infection and it is also home to many migrant factory workers. In Nepal, we selected two settings: Chandragiri municipality of Kathmandu District (Bagmati Province) and Gulariya municipality of Bardiya District (Lumbini Province). These districts have geographical, social and cultural variations and both host the provincial capital cities (Kathmandu is a hill district; Bardiya is a terai district bordering with India). In Sierra Leone, two districts were selected: Bonthe district in the Southern Province, a hard-to-reach area, which is riverine with several islands, with strong donor support for the CTC programme; and Kenema district in the Eastern Province, with large urban and rural areas, with minimal support from partners for the CTC programme. Data collection was conducted between December 2020 and February 2021.

### 2.1. Document Review

We reviewed literature on the gendered experience of CTC providers during the COVID-19 pandemic in Lebanon, Nepal, Myanmar, and Sierra Leone from searching electronic databases (such as CINAHL, Medline, Google Scholar), websites of national and international NGOs and networks websites (such as WHO, UNICEF) and government departments involved in the COVID-19 response such as the Ministry of Health for relevant reports, commentaries, working papers, policy documents, guidelines and training materials. We also searched the reference lists of all included documents to identify additional relevant documents.

### 2.2. Key Informant Interviews

Using piloted topic guides we explored local stakeholders' perceptions of CTC providers' gendered roles and work in the COVID-19 response, and the support that was provided. We selected key informants who were knowledgeable about the work of CTC providers including managers of health facilities who supervised CTC providers and community and district level supervisors (Table 1). In Lebanon and Myanmar, the interviews

were conducted using telephone or WhatsApp because of the COVID restrictions in travel. In Sierra Leone and Nepal, the interviews were carried out face-to-face in offices or homes, while applying local COVID-19 public health measures. The experienced qualitative research teams conducted the interviews in the local language in each country (Lebanon—Arabic; Myanmar—Burmese; Nepal—Nepali; and Sierra Leone—Krio). The interviews lasted between 35 and 90 min and were recorded following consent of the participants.

**Table 1.** Details of the key informants.

| | Site 1 | Site 2 | Total |
|---|---|---|---|
| Lebanon | 3 managers of facilities (3M) | 3 managers of organisations providing outreach (3F) | 6 (3F;3M) |
| Myanmar | 2 township supervisors (1M; 1F) | Not applicable | 2 (1F;1M) |
| Nepal | 1 Sub-health coordinator (COVID-focal person) (M) <br> 1 Ward chair (F) <br> 1 Mayor (M) <br> 1 Health Post In-charge (M) | 1 Health Post In-charge (F) <br> 1 Health Post In-charge (M) <br> 1 Health Coordinator (M) <br> 1 Mayor (M) <br> 1 Public Health Inspector (COVID focal person) (M) | 9 (2F; 7M) |
| Sierra Leone | 1 District health Management Team member (M) <br> 1 CTC Peer Supervisor (M) <br> 1 Section Chief (M) <br> 1 Mammy Queen (Chairlady) (F) | 1 District health Management Team member (M) <br> 1 CTC Peer Supervisor (F) <br> 1 Section Chief (M) <br> 1 Mammy Queen (Chairlady) (F) | 8 (3F; 5M) |
| Total | 13 | 12 | 25 (9F;16M) |

Key: F = female; M = male.

*2.3. In-Depth Interviews or Focus Group Discussions with CTC Providers*

Using piloted topic guides we explored CTC providers' roles, interactions with health systems and communities, support received, and the challenges they faced during COVID-19, and how these are affected by gender. In Lebanon, Myanmar and Sierra Leone we included male and female CTC providers who were working during the pandemic. In Nepal, we focused on Female Community Health Volunteers (FCHVs). In Lebanon, Myanmar and Nepal, it was not possible to bring CTC providers together due to COVID-19-related restrictions, so we conducted individual interviews. In Lebanon and Myanmar these were via telephone or WhatsApp, whereas in Nepal these were face-to-face. In Sierra Leone, national guidelines allowed CTC providers to come together, and we conducted focus group discussions (FGDs) applying local public health measures to minimise transmission of COVID-19 (Table 2). The research teams in each country conducted the interviews or discussions in the local language. The interviews lasted between 35 and 65 min and the FGDs lasted between 120 and 150 min.

**Table 2.** Details of the CTC providers.

| | Site 1 | Site 2 | Total |
|---|---|---|---|
| Lebanon | 6 CTC providers (4F;2M) | 6 CTC providers (4F;2M) | 12 (8F;4M) |
| Myanmar | 5 CTC providers (4F;1M) | Not applicable | 5 (4F;1M) |
| Nepal | 7 FCHVs | 6 FCHVs | 13 (13F) |
| Sierra Leone | 1 FGD with women CTC providers (8) <br> 1 FGD with men CTC providers (8) | 1 FGD with women CTC providers (8) <br> 1 FGD with men CTC providers (7) | 31 (16F;15M) |
| Total | 34 (23F;11M) | 27 (18F;9M) | 61 (41F;20M) |

Key: F = female; M = male.

*2.4. Analysis*

The research team transcribed verbatim the interviews and FGD recordings and translated them into English for analysis. We analysed the document review and qualitative data using the thematic framework analysis approach (Ritchie et al. 2003), supported by NVivo 11 software. We developed a coding framework for all settings developed from the topic guides, research objectives, themes emerging from reading the transcripts and data whilst being informed by Steege et al.'s (2018) community health worker and gender framework and Morgan et al.'s (2016) gender analysis framework. The research teams applied the framework to the transcripts and data and developed charts for each code. The research team then identified and agreed key themes across the contexts through virtual workshops and e-mail exchange.

**3. Results**

The results begin with an overview of the findings from the document review. We then move on to themes emerging from the interviews and FGDs with CTC providers and key informants framed by the following questions guided by Morgan et al.'s framework: Who has what? Who does what? How are values defined? Who decides? How is power negotiated and shaped by people and environment? Sub-themes under each question are included and this structure is used to present the qualitative results (Figure 1).

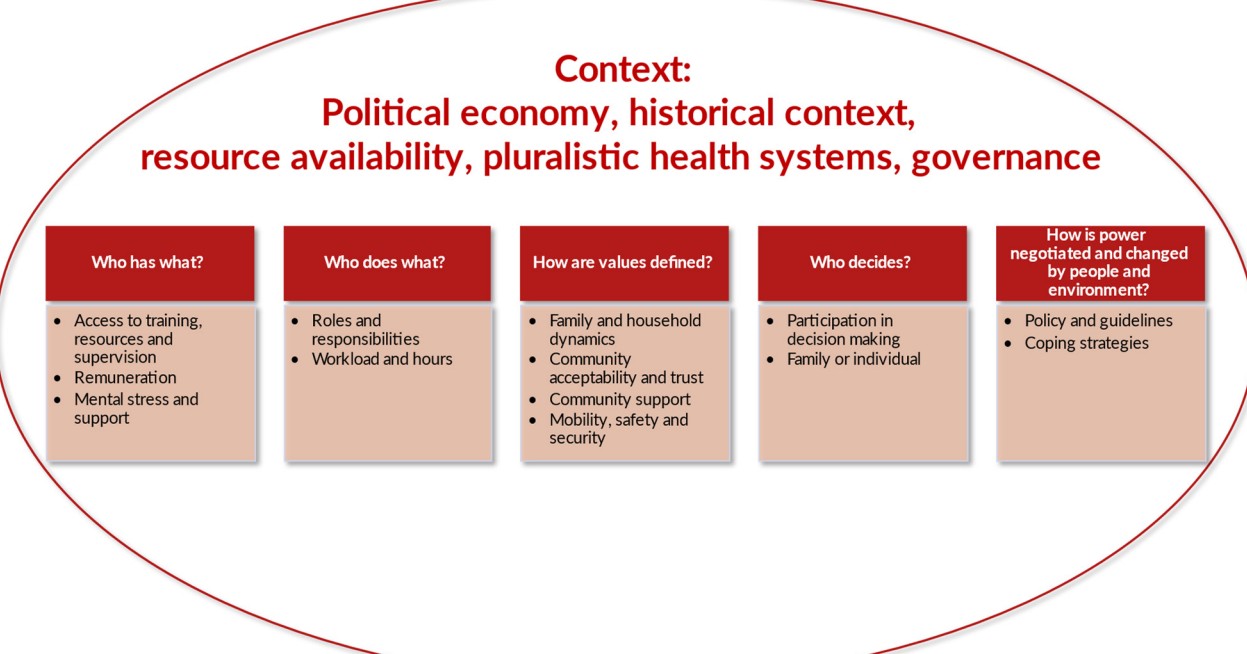

**Figure 1.** Gender analysis framework (adapted from Morgan et al. 2016).

*3.1. Document Review Findings*

Table 3 provides an overview of the documents reviewed. There was little information in the documents reviewed about the gendered experience of CTC providers during the COVID-19 pandemic in the study settings. Some documents described the roles and responsibilities of CTC providers in the COVID-19 response, including awareness raising about COVID-19, prevention and treatment, contact tracing and monitoring people in quarantine (Government of Sierra Leone 2020; El Chammay and Roberts 2020; UNHCR 2020; WHO EMRO 2020; Parajuli et al. 2020; Aryal and Pant 2020; International Organization for Migration 2020; United Nations Foundation 2020; Ministry of Health and Population

Nepal 2020). None of the documents reported differences in the roles and responsibilities between women and men CTC providers.

**Table 3.** Documents reviewed in each country.

|  | No. Documents Reviewed | Types of Documents |
|---|---|---|
| Lebanon | 59 | - Governmental reports and web pages: 3<br>- UN and NGO reports: 33<br>- Blogs: 2<br>- News articles: 16<br>- Peer reviewed journal publications: 5 |
| Myanmar | 7 | - Government policy: 1<br>- Peer reviewed journal publication: 1<br>- NGO blogs: 1<br>- News articles: 4 |
| Nepal | 16 | - Peer reviewed journal publications: 5<br>- NGO documents: 7<br>- Government policies: 4 |
| Sierra Leone | 6 | - Peer reviewed journal publications: 1<br>- NGO blogs: 4<br>- Government policies: 1 |

Some documents explained why CTC providers are important cadres in the response including their ability to communicate easily with community members as they speak local languages and understand the community norms, and are often trusted by the community (Government of Sierra Leone 2020; Parajuli et al. 2020; Akseer et al. 2020). Challenges faced by CTC providers were described in some documents, such as transport, community stigma and support from families but were generally not differentiated by gender (Kanu et al. 2021; Mutseyekwa 2020; Partners in Health 2020; Parajuli et al. 2020; Nepal and Aryal 2020; Public Services International 2020; Menge and Paudel 2020; Aryal and Pant 2020). However, one document in Myanmar acknowledged that women CTC providers have dual responsibilities of household/family and community work (Lambrecht et al. 2020). There were guidance documents for CTC providers working in COVID-19 response in Lebanon and Sierra Leone (Government of Sierra Leone 2020; Ministry of Public Health Lebanon 2020, n.d.; World Health Organization 2020; Ayadi et al. 2020). However, these did not include any specific support or guidance for women and men CTC providers. This document review highlighted an important evidence gap. The following section provides detailed findings from the four study settings on the gendered experiences of CTC providers during COVID-19.

*3.2. Findings from the KIIs, IDIs and FGDs*

3.2.1. Who Has What?

Access to Training, Resources and Supervision

In all contexts there was limited training and supervision to work effectively during COVID-19. In Nepal, Lebanon and Sierra Leone, some training was provided which helped CTC providers understand more about COVID-19, how to use face masks, and physical distancing. In Lebanon, training was reinforced through online reminders. However, in Myanmar and Nepal, CTC providers relied on other sources of information and support, such as radio, TV, and Facebook, and quick updates via phone.

> "There was no proper training—our main source of information was Facebook".
> CTC provider, woman, Myanmar, IDI

Additionally, in Nepal, CTC providers also reported to and received updates from their health facility managers via telephone. After the lockdown was lifted, regular

monthly meetings at health posts were used as a platform to share and learn any new information about COVID-19.

> "It was one way communications. When the phone rang, we just had to receive. Then, they would provide all the information about the symptoms. Only when they ask us to guess the right answer, we would guess it. Then they would respond that our answer is correct. That's all". CTC provider, woman, Nepal, IDI

In all contexts, there was limited equipment and supplies with Personal Protective Equipment (PPE) and hand sanitisers often running out; CTC providers then either reused them or bought additional supplies using their own money. There was a shortage of PPE in all settings and particularly in the first wave of the pandemic, and priority was given to health workers in health facilities and in Nepal, CTC providers involved in contact tracing.

> "We can hardly secure the PPE for the employees". Man, Lebanon, KII

In Myanmar, CTC providers reported no support or acknowledgement for their work from the Township Health Department. However, they mentioned receiving some support such as hand gel and mask provision from local NGOs. They explained that if they had been trained and given information, their work would have been more effective. They all felt that the Township Health Department was not organized and needed to take more responsibility.

> "It would be really good if the health department recognizes the community health volunteers and tries to connect and collaborate with them more". CTC provider, woman, Myanmar, IDI

In Myanmar, CTC providers reported that they were not prioritised for COVID-19 vaccination. In the other settings, CTC providers did not mention vaccinations perhaps because the COVID-19 vaccination roll out had not started in Lebanon, Nepal or Sierra Leone at the time of data collection.

Remuneration

There was limited financial support for additional COVID-19 roles undertaken by the CTC providers in all settings. In Myanmar, CTC providers paid out of their own pocket for transport when making home visits and telephone bills when communicating with women and families. In Lebanon, CTC providers and managers did not report any direct gender differences in the allocation of salaries or bonuses. However, men were more likely to be assigned to higher-level posts with more responsibilities and better pay because of their advanced educational background, community preference for men in positions of leadership and that they were more able to accept overtime work with its extra pay. In contrast, women rushed home to take care of their families.

In Nepal and Sierra Leone, government guidelines stated that CTC providers should be given additional incentives: the equivalent of $20 per week for COVID-19 activities in Sierra Leone; and Rs. 1500 per day (or US$12) in Nepal. However, many CTC providers did not know about this, and did not receive the benefit. Where they did, they perceived it as too small compared with their workload and the associated level of responsibility and risk.

> "The money that they gave us was too small . . . during a lockdown, we have to buy things in the house to eat. The family burden is too much on us we have our children to look after and other family members". CTC provider, woman, Sierra Leone, FGD

In Sierra Leone, during the pandemic, lockdown, travel restrictions and increased workload hindered additional income-generating activities which CTC providers usually undertake to support their families. The price of food and other staple goods increased dramatically during this time due to travel restrictions, and this increased the stress CTC providers experienced. It was particularly challenging for women CTC providers who were widows or single parents:

> "It was not easy . . . especially for us with kids . . . we were just working in the
> community and we had nothing . . . so the little we had is the only thing we were
> using". CTC provider, woman, Sierra Leone, FGD

Mental Stress and Support

CTC providers in all settings reported increased stress during COVID-19. This was because of fear of contracting and passing COVID-19 infection on to relatives, limited provision of equipment, inadequate information and being stigmatised by the community. As one CTC provider in Sierra Leone explained:

> "We were bashed at in some communities . . . sometimes we cannot even do our
> work . . . as most were claiming that we were disease carriers . . . coupled with a
> lot of provocation so that led to serious mental health issues amongst us". CTC
> provider, man, Sierra Leone, FGD

In Lebanon, many respondents said that working in the COVID-19 response brought more stress to women than men, due to the pressure of both work and home responsibilities. To cope with this psychological distress, CTC providers doubled their efforts to respect personal safety measures, relied on family support, and a few used professional counselling services available through online consultations.

There were no specific or formal support mechanisms in place for CTC providers in Nepal and Myanmar. In Sierra Leone, despite complaints to their managers, CTC providers did not receive any mental health support, even though there is one mental health nurse in each district. They were advised by their managers to ignore criticisms from community members and continue doing their work.

> "We forwarded several complaints to our superiors . . . we were admonished to
> give deaf ears and do the job as we have decided to help our communities . . .
> Our superiors also had engagements with community leaders, so that they can
> take measures as safeguards for responders to do their work effectively". CTC
> provider, Man, Sierra Leone, FGD

This limited mental health support for CTC providers reflects the shortage of formal mental health services available in these contexts. Instead, CTC providers relied upon moral support from their supervisors or from their families. In Myanmar all CTC providers expressed that they were "givers" of moral and mental support to community people rather than "receivers".

### 3.2.2. Who Does What?
Roles and Responsibilities

There were reported gender differences in roles in some settings. In Lebanon, men CTC providers are trusted with assignments with more responsibility, with women holding roles as health educators. Similarly, in Myanmar, men CTC providers take administrative roles, whilst women focus on health education and COVID-19 prevention activities.

In addition, when the pandemic started, in Myanmar, Nepal and Lebanon many of the CTC providers did not start COVID-19 work immediately. In Myanmar and Nepal, CTC providers provided advice and support to their community members by telephone, often being the first to support the community. In Lebanon the health centres from which the CTC providers operated were closed for a few months, and then gradually reopened. In Sierra Leone CTC providers had COVID-19-specific roles allocated to them at the start of the response.

CTC providers experienced a "layering" of responsibilities. They had their usual responsibilities of general health education, maternal and child health care, and clinical work; and on top of this they had the COVID-19 responsibilities such as awareness raising about hygiene and physical distancing, contact tracing, screening, referral of people with COVID-19, attending to people who were in home quarantine, and providing psycho-social sup-

port. Figure 2 presents the many roles and responsibilities of CTC providers, and also the reasons why they took on these roles during COVID-19.

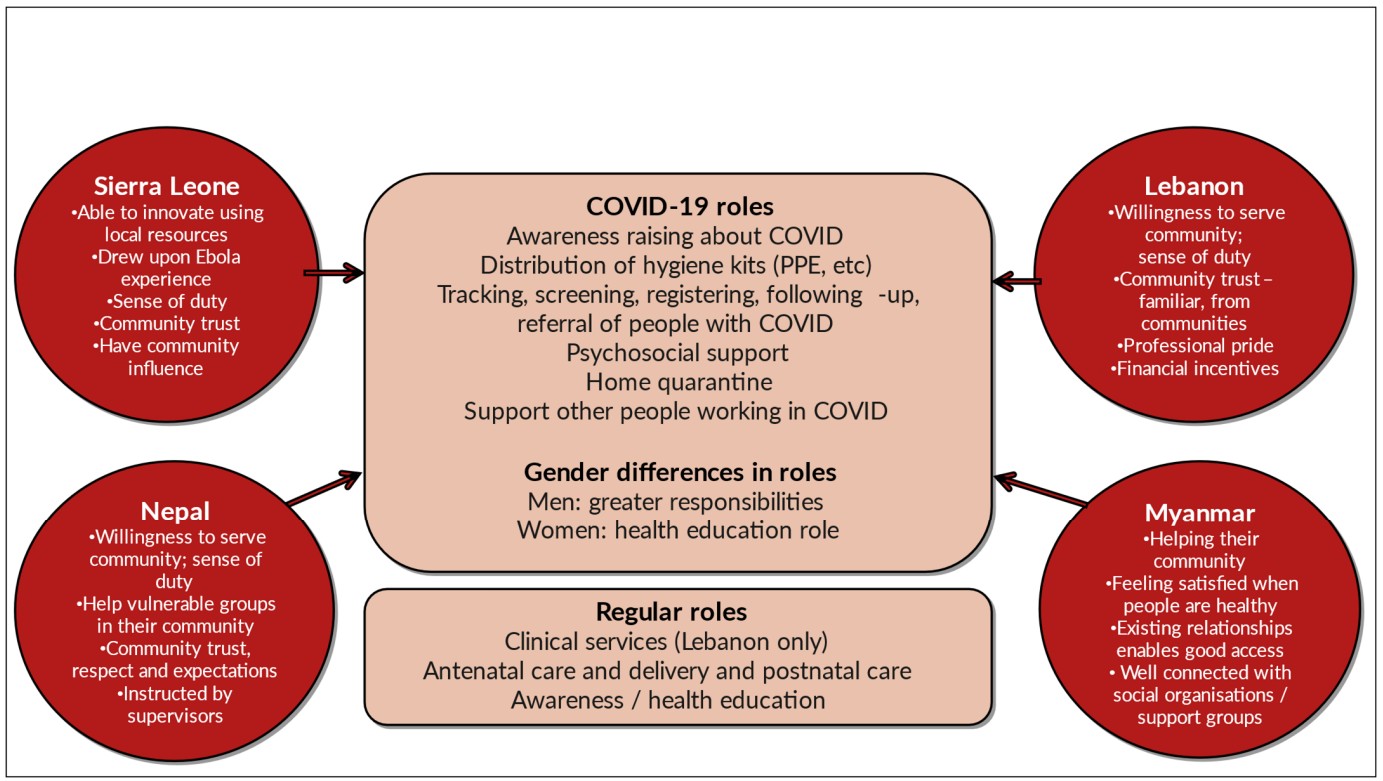

**Figure 2.** Adapting roles and responsibilities of CTC providers during COVID-19.

Workload and Hours

There was increased workload for CTC providers in all settings. For example, in Sierra Leone, CTC providers were expected to provide twice daily reports on families or individuals in quarantine, as one explained:

> "The workload was much . . . we have to visit quarantine homes twice a day and talk to them and check their temperature . . . And in some households, there are many people". CTC provider, woman, Sierra Leone, FGD

All CTC providers were managing multiple tasks. However, community gender norms in all settings meant that women CTC providers faced having to juggle multiple roles, including regular community work, COVID-19 activities as well as housework, looking after families and home-schooling during lockdown, bringing additional pressure and stress. It was especially challenging when additional work came up unplanned—women CTC providers found this particularly difficult as they were trying to manage their homes and families too.

> "Need to adjust to have equilibrium between work and family. More sacrifice for women than men as men do not have much responsibility like women". CTC provider, woman, Myanmar, IDI

> "Family comes first: When the children are not going to schools, it exerts pressure on me. I am working outside, and when my children study online, I had to stay with them for hours. My husband is at home due to corona. This also exerts pressure". CTC provider, woman, Lebanon, IDI

> "They gave me a phone call from health post instructing me to deliver kit box to COVID patients and bringing back their health reports. Both works [farm

work and duty as CHW] coincided at the same time. There was no one at home. I asked my neighbour to prepare some pickle and snacks for farm worker. I cooked potato and handed it to her. After I reached health post to receive kit box, I called 2, 3 people from my own community and sent them to the field. Then after, I visited to COVID patient's home". CTC provider, woman, Nepal, IDI

### 3.2.3. How Are Values Defined?

Family and Household Dynamics

Family support was important for CTC providers in all settings to be able to fulfil their work during COVID-19. This was particularly important for women CTC providers. For example, in Nepal, family members assisted by completing household chores (cooking food, taking care of children). In Myanmar, families helped with organising transport, accompanying CTC providers as they did their work, and advising them to take extra precautions to prevent infection. In Sierra Leone, relatives gave words of advice and encouragement.

"My family and friends supported me and they encourage us to do the work . . . They listen to whatever we tell them to do that alone is a big support, because they are making our work easy". CTC provider, woman, Sierra Leone, FGD

However, this family support varied amongst the CTC providers. The support depended on how family members valued community work, their views on voluntary unpaid work, other sources of income for the family, as well as how it interfered with household work. They were also concerned about exposure to COVID-19 and the CTC providers becoming ill and passing the virus to their family.

Some CTC providers were able to cope with this by waking early to complete their household chores before starting their community work; and recognising the value of their work and ignoring the reprimands. CTC providers in Nepal explained that despite supervisors knowing these challenges, there was no support from the health system.

"The family member scold asking, "Why do you need to do that work?" My own husband scolds me asking, "How much do they pay you? Why do you go there?" I have been used to this type of scolding. I let him continue scolding. I'll carry on with my work". CTC provider, woman, Nepal, IDI

Community Acceptability and Trust

Community trust and acceptability enabled CTC providers to take on COVID-19 work. Most CTC providers were trusted by the communities—they had developed relationships built up through their community work, and from living within the communities. They could draw upon existing links with community leaders and organizations to do their work. They were willing to take on these roles and saw it as their duty in a time of emergency. In Sierra Leone, CTC providers drew upon their experience of working with communities during the Ebola Outbreak, to talk with people and reduce fear and anxiety. The trust that they had developed during their work in the Ebola crisis helped them to work with community members during the COVID-19 emergency.

However, there were some examples of stigma and discrimination towards CTC providers and their families in all settings. Community members were reluctant to listen to their advice and health messages and receive care; they were also isolated from any social activities as they were viewed as carriers of the virus.

"Neighbours and people around me used to tell me "Stay away from us. Don't come closer. You work with Corona. Maybe you could infect us"." CTC provider, woman, Lebanon, IDI

"Some of them lock their door. When we rang bell, they used to look at us and go inside fearing whether we might have brought corona. When we told them, "We need to discuss something with you," only then they came out to their balcony". CTC provider, woman, Nepal, IDI

In Sierra Leone, some CTC providers were accused of prolonging COVID-19 so that they could receive more money, as the community perceived that CTC providers were receiving large financial rewards.

> "[Community] are assuming that we are also playing a huge part in prolonging the period of the disease in communities ... In communities, we are only regarded by old people ... while young people are accusing us of monetizing the whole response programme ... that creates a kind of stigma around us in our communities" CTC provider, Man, Sierra Leone, FGD

Gender norms influenced women CTC providers' experiences of providing support, advice and care in their communities. CTC providers and key informants in Sierra Leone, Myanmar and Nepal reported that there was some reluctance to listen to women CTC providers. Women CTC providers struggled to convey messages and advice to some men in the communities. In Sierra Leone, some supervisors paired women with men CHWs to overcome this issue.

> "Communities do not tend to often listen to women in certain situations due to cultural beliefs ... they are not given the audience they need ... so in some cases we provide them with a male back up if there should be pressing issues to be addressed". Man, Sierra Leone, KII

The stigma and discrimination appeared to reduce as the pandemic continued for example in Myanmar stigma was observed in the early phase of the COVID-19 pandemic as the community feared contracting the disease. However, as wearing masks and hand washing increased, less stigma was seen in the second wave, as one woman explained:

> "Initially during the first wave some people in the community were frightened to talk to me which eventually reduced in the second wave". CTC provider, woman, Myanmar, IDI

CTC providers were able to reduce this stigma and discrimination by talking with community members, drawing on the trust they had gained before the pandemic.

Community Support

There were several examples of organised support from communities for CTC providers. In Myanmar, the township already had a very supportive community structure, and established a Township Coordination Committee for COVID-19 where community-based charity and social organizations came together and showed appreciation for the CTC providers' work. CTC providers were also part of the ward community groups which were very active in supporting CTC providers in identifying and referring people with COVID-19 and their contacts for treatment or quarantine.

> "Tokens of appreciation and thank you cards from the Myanmar Red Cross Society, community and health departments make us feel motivated. I feel like I was supported when I felt very tired. I am happy for that. We work not for money and not expecting we will get something in return. From MRCS, we got hats or t-shirts and some other presents from community as tokens of appreciation". CTC provider, woman, Myanmar, IDI

In Sierra Leone, community leaders and other community stakeholders instituted by-laws that supported CHWs to work outside of their home communities during the response, helping their entrance to, and acceptance by, the community. Existing support groups, such as men's and women's support groups, mediation bodies between CHWs and community members, and Village Development Committees, were also reported as being supportive during the response. NGO partner support was also reported in Bonthe district, as one CTC provider explained:

> "After realizing the intensity of the outbreak, we started getting support from some other local NGOs ... materials like thermometers and hand washing ma-

terials were provided to us . . . we distributed those at strategic points that could be easily accessed by people to use". CTC provider, man, Sierra Leone, FGD

In Nepal, locally elected representatives, members of the Citizen Youth's group, and health workers from the health posts supported FCHVs by requesting community members to listen to the advice being given by the FCHVs.

In Myanmar, Sierra Leone and Nepal, community structures were already established and working, and these existing relationships with communities, health facilities and NGOs were drawn upon to support CTC providers' work. In Lebanon, these community structures are less evident and CTC providers did not report strong community support.

Mobility, Safety and Security

Challenges in travelling to and from work safely were raised in all settings, with strong gendered implications. In Myanmar, women experienced restrictions in travelling especially at night. Women CTC providers needed to be accompanied and relied on their families to do this.

"There is a challenge for girls, and it sometimes needs the family to accompany the girls (volunteers) on their way home from work. Sometimes, we have meetings at night. For me, my husband come and pick me up. For girls, we may need to arrange for their return trip. For example, the township committee arranges a car for girls. For boys, there is no problem as they can manage on their own". Woman, Myanmar, KII

In Lebanon, public transport is unsafe for women, with risk of assault being very high, especially in taxis. Women travelling alone, especially at night, is frowned upon:

"Their thinking is like: you are a girl and you are attending camps by yourself. People ask: you have nobody else? And even when we quit the camp and look for a car, we are not viewed positively". CTC provider, woman, Lebanon, IDI

3.2.4. Who Decides?

Participation in Decision Making

CTC providers in Myanmar reported that most decisions about COVID-19 activities were made by the Township COVID-19 Coordination Committee. CTC providers explained that men have more freedom than women when choosing timing, location and type of work they do, as they have fewer family responsibilities and more sense of security.

In Nepal, many CTC providers described feeling empowered because of the work that they do that provides opportunities to enhance knowledge and skills, be connected with the outside world, and contribute to improving the lives of vulnerable populations. However, their responsibilities were not clearly explained nor were they involved in the decision-making process about their mobilisation during the pandemic. This was shown in one study setting, where COVID-19 prevention and control groups were established in the municipality in line with national policy guidelines. These guidelines stated that at least one local volunteer CTC provider should be in these groups. However, CTC providers were unaware of these groups:

"No, I am not aware of COVID-19 specific group formation in the community, and I don't know about my membership on it either". CTC provider, woman, Nepal, IDI

In addition, men held most of the decision-making roles such as Representative of Ward Offices and In-Charges of health posts. Some CTC providers reported that planning and implementation process are not always gender-sensitive, and the specific needs of women are not always addressed. They explained that if the leaders were women, the planning and implementation would have been more gender sensitive:

"Yesterday, there was a certain programme led by Ward Officials. I didn't know about the programme. A Ward member invited me at the last hour. They

should've timely informed me about the program in the morning or in the afternoon so that I could get time to manage my household chores". CTC provider, woman, Nepal, IDI

In Sierra Leone, CTC providers felt that all decision making was done at facility, district or central level, with little if any discussion with them. Some explained that they do not know where to report because they are unsure who makes decisions about their affairs including equipment and supplies and other administrative issues they may face.

In Lebanon, CTC providers reported that they made decisions about participating in the COVID-19 response, managing their work, and were consulted on other decisions such as stopping outreach visits, limiting socialization between colleagues inside the facility, and putting in place strict hygiene measures at the peak of the epidemic. One woman manager explained:

> "If they have children but don't have problem with working as frontline workers within COVID-19, it is their own decision. We did not oblige anyone to participate in the response ... We also leave the choice for the person. Also, when we have emergency cases, we need to stay late, the person who wants to leave could leave. We don't exert any retaliation [penalties]". Woman, Lebanon, KII

However, one CTC provider said that she had signed a contract before COVID-19 that requires her to accept any mission assigned by the organization.

Family or Individual Decision Making

In Myanmar, women CTC providers reported making their own decisions about their work, but often consulted their husbands for advice. In Nepal, family support was critical to CTC providers taking on COVID-19 work. Inadequate supply of safety items led to family members stopping CTC providers working in the communities during the pandemic, due to fear of transmission of COVID-19 to family members. Although such issues were raised and discussed with supervisors during meetings, these were not addressed. In Lebanon, CTC providers, especially women, described great changes in their family life due to COVID-19 and the lockdown measures, such as increased anxiety of children and husbands, poverty due to loss of employment, and improved communication within the family, which affected how they made decisions about work. In Sierra Leone, some CTC providers received family support in the form of encouragement. At the household level, the outbreak affected family dynamics especially with the lockdown of school. Gender norms dictated that mothers had to stay home to look after the children, despite having to execute their roles as CTC providers and domestic duties. One male CTC provider expressed concern over the likelihood of teenage pregnancy during this period, and the financial implications on him as the breadwinner in having to look after a teenage mother and her baby.

> "I was really traumatized due to the pregnancy of my younger sister ... life was so stressful for me due to additional burden ... I have to take care of both the mother and the child ... My job was affected due to stress ... no help from anywhere else ... I have now got more burden on me with less income". CTC provider, man, Sierra Leone, FGD

3.2.5. How Is Power Negotiated and Changed by People and Environment?
Policy and Guidelines

CTC providers in Myanmar were not aware of any policies and did not receive any guidelines for managing COVID-19. Some received leaflets to distribute in their community. In Nepal, there were frequent changes in policy and guidelines as understanding of COVID-19 rapidly evolved. These frequent changes, delayed policy and guidelines communication and resulted in patchy information to CTC providers.

In Sierra Leone, manuals, guidelines, posters, wall posters and booklets containing information about community sensitization on prevention of COVID-19 were provided.

> "During the training at the council, they gave us posters that we used to sensitize people . . . telling people about the prevention of COVID-19, they gave us megaphones to do the sensitization". CTC provider, woman, Sierra Leone, FGD

In Lebanon, all respondents reported that they have full access to guidelines and instructions concerning precautionary measures and personal safety pertaining to COVID-19. In addition, hygiene and personal safety instructions were regularly delivered through WhatsApp as one CTC provider said:

> "We have a protocol to approach patients with maximum levels of protection for ourselves, the patients, and the society". CTC provider, man, Lebanon, IDI

Coping Strategies

Many CTC providers relied on their intrinsic motivation to carry on working in these difficult times. For some, religion was important, for others, trying to look at the positive side was critical.

> "Every morning, I say to myself, "You . . . are a strong person. You need to bear the difficulties thrown in your way, and you have to challenge them . . . " Thanks to God, I have work, even if it is tiring, but not every refugee has the chance to work". CTC provider, woman, Lebanon, IDI

To cope with stress and anxiety they often spoke with their families, peers and sometimes with their managers. As one woman CTC provider in Lebanon explained:

> "I come and talk to my husband. I tell him about my day, and he calms me a bit. Usually, each family member needs to be giving to each other in order to build the family". CTC provider, Lebanon, IDI

In Nepal, CTC providers developed a routine to accomplish both household and professional responsibilities. For instance, waking up early to complete the household chores before going into the community. They also reported extensive measures to keep their family safe, such as staying on a different floor in the home, washing hands, showering and changing clothes after coming home, reusing masks after washing them, or paying out of their own pocket for additional masks when their supply had run out. They felt demotivated that despite their commitment and hard work, the health system did not support them.

> "How long will the 15 masks last for? [Laughs] After that, I have been buying them myself until now. My husband buys them in a packet. Everyone uses them. Mostly, we throw them after returning home. We also wash them and reuse sometimes. How often can we buy then? We get tired of buying them continuously. Moreover, it is said that the cloth masks aren't appropriate. Hence, I reused them". CTC provider, woman, Nepal, IDI

## 4. Discussion

### *4.1. Summary of Key Findings*

Our study has focused on the roles of CTC providers and their gendered experiences during the COVID-19 pandemic and in the often-neglected contexts of fragile settings, using a gender analysis framework to present our findings (Morgan et al. 2016). We now synthesise our findings against the conceptual framework of Steege et al. (2018). This framework shows how gender roles, relations and norms shape CTC provider experience at the individual, community, and health system levels, the relationship between them, and the complex interplay within the broader health and political environment. We have adapted this framework (in italics in Figure 3) as our findings highlight the unique and added value of researching and understanding the gendered experiences of CTC providers in different fragile settings during COVID-19 and assessing implications for gender equity and justice. Our findings are relevant to other emergency situations and crises, which test fragile health systems.

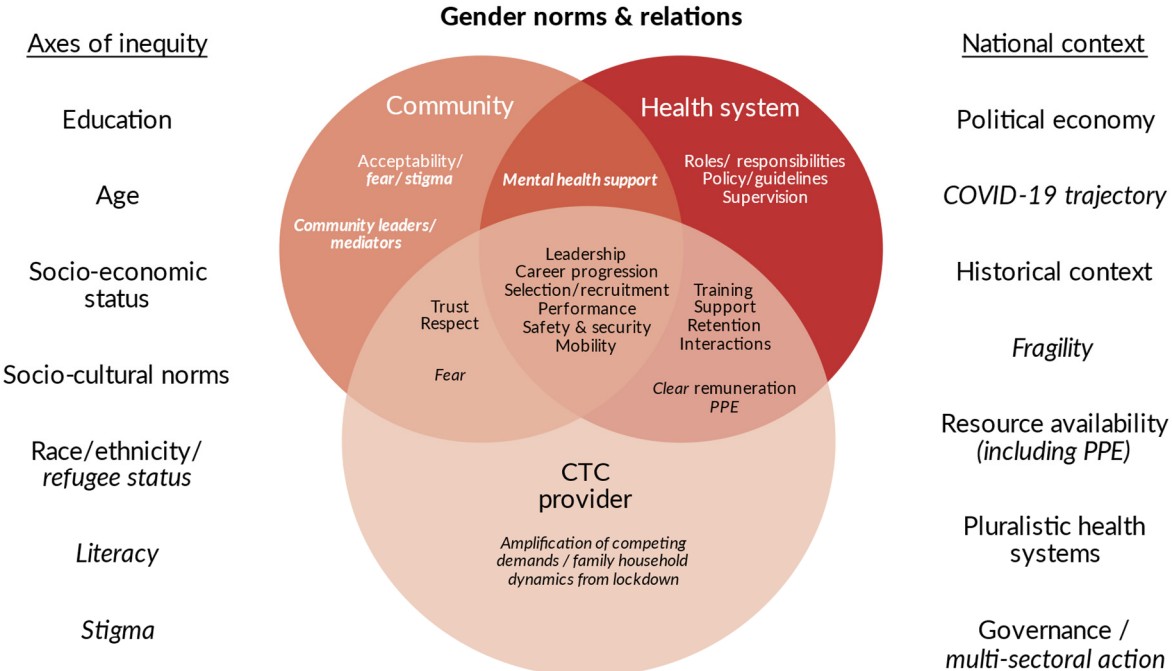

**Figure 3.** Gendered experiences of CTC providers in fragile settings during COVID-19 (adapted from Steege et al. 2018). Key: italics—additions from this study.

For each area in Figure 3—CTC provider and family, community and health system—we summarise what we have learned and how this relates to existing literature, and the way forward in terms of implications for practice and policy.

*4.2. CTC Providers and Families*

Our study illustrates that during COVID-19, there was an amplification of competing demands on CTC providers, including effects on household and family dynamics during lockdown and travel restrictions. The COVID-19 pandemic with lockdowns and school closures has brought additional stresses for families worldwide, with a gendered division of labour meaning this is acutely felt by women (EMERGE 2020). There is a large body of literature collated by the Gender and COVID-19 Project that conducts gender analysis to identify and document the gendered dynamics of COVID-19 (see website https://www.genderandcovid-19.org/ accessed on 15 February 2022). For example: a literature review identified that there is strong evidence of the disproportionate increase in unpaid domestic burdens for women relative to men (EMERGE 2020). Studies have identified that lockdowns increased women's burden of unpaid work in India, Australia, Turkey (Chauhan 2020; Craig and Churchill 2020; İlkkaracan and Memiş 2020). CTC providers' experiences in this study also reflect these patterns and these gendered stresses are magnified by: the role being poorly remunerated or not remunerated at all, and CTC providers and family members seeing this work as being risky.

CTC providers need family support to do their work which includes an understanding of the critical role that they play during emergencies such as COVID-19, help with household chores, and support with safe and reliable transport. This could be stimulated through more communication with families and greater public recognition from the health system and community for their work. However, reliance on family members for safe transport needs to be reconsidered, with the health system providing more support in this area.

### 4.3. Community

Our study demonstrates that CTC providers still largely enjoy trust and are embedded within their communities with a strong willingness to serve. However, in some cases, settings or moments of time within the evolution of the pandemic CTC providers experienced fractures in this community trust and were viewed as carriers of the virus and the way in which this stigma was experienced was gendered. This resonates with findings from other settings such as Brazil where community health workers experienced threats and aggression (Lotta et al. 2020) and in India where harassment and disrespect fuelled by fears of COVID-19 have been reported (Nanda et al. 2020). Community health workers in previous pandemics experienced stigmatisation, isolation and were socially ostracised (Bhaumik et al. 2020). Other ramifications were evident in our study, for example, in Sierra Leone, CTC providers being accused of prolonging COVID-19 so that they could receive more money. Health systems actors such as policy makers and managers often rely on CTC providers' individual relationships with community members to enable them to carry out their health activities. However, community and traditional leaders and organisations have a role to play in supporting CTC providers, and this support could be strengthened. For example, community and traditional leaders can openly support CTC providers' work at community meetings and through other communication mechanisms (LeBan et al. 2021), and mobilise resources such as transport, food or assistance with childcare or food production (Caperon et al. 2021). Clear messaging about CTC providers' roles in the community by the community leaders and health system actors is needed as well as strategies to address stigma.

### 4.4. Health System

Our study showed that CTC providers experienced a layering of roles and responsibilities on top of their routine work and family commitments, with gender differences in how this plays out which are linked to existing gender norms. This is in keeping with other research that shows CTC providers' workloads stretching across multiple vertical programmes (Raven et al. 2015; Oluwole et al. 2019; Kok et al. 2017). Our study also demonstrated that they did not always have the support that they needed to undertake the challenging role of providing support and care about COVID-19 to their communities e.g., supervision and support, financial incentives, mental health support, PPE and training and information. In all contexts, there are issues with provision of resources and support to CTC providers prior to COVID-19. For example, CTC providers would usually receive some training and supervision related to their work, although the frequency and quality of this varies across different contexts. Mental health services are limited in all settings for all health workers, including CTC providers. COVID-19 has shone a spotlight on these resource gaps, and highlighted the need for robust and responsive systems. Other axes of inequity, such as the refugee status of CTC providers in Lebanon or the low-literacy of some CTC providers hindered their ability to perform the work and affected their experiences. In the national context, there were several additional factors that influenced the experience of CTC providers including:

- The pandemic trajectory, the magnitude and speed of infection.
- The fragility of the country and how quickly and well the country and health system could respond.
- The resource availability and in particular the ability to provide PPE to health workers including CTC providers, and how this was prioritised.
- The need for multisectoral action in the COVID-19 response, and how to support CTC providers was identified.

Managers of CTC providers and those responsible for developing policies and guidelines related to CTC providers can strengthen training, access to information and supervision so that CTC providers have up-to-date, accurate and culturally appropriate information. Developing training or guidance documents that can be provided through mobile phones could help here. Supervision from peers or managers through mobile platforms

such as WhatsApp can support CTC providers, especially in remote areas or where travel is unsafe (O'Donovan et al. 2021). The incentive policy needs to be transparent and followed so that this does not damage CTC providers' relationships with the community as seen in Sierra Leone. Adequate equipment and supplies to protect CTC providers from infection generates confidence in their ability to provide services safely and can build family support for their work; and ways to ensure that CTC providers are not forgotten when it comes to provision of PPE and vaccination.

Support for the mental health of CTC providers is long overdue, and the COVID-19 pandemic has amplified this need (Dean et al. 2020; George Institute 2020; Yakubu et al. 2022; UNICEF, Save the Children, and International Rescue Committee 2020). Investment in caring for the mental health of CTC providers such as mental health first aid training, peer support, and pathways for professional support is urgently needed.

Finally involving CTC providers, and in particular women, in planning and decision-making processes when emergencies happen is critical for effective community-based interventions, as they have the knowledge and understanding of communities, as described in a recent analysis of the gender, equity and justice issues surrounding health workers in fragile settings (Mansour et al. 2022).

*4.5. Strengths and Limitations*

There are several strengths in this study. We had experienced and embedded qualitative research teams in each context who were used to working with community level health staff. The teams worked together to support each other through the research process and analysis and shared their perspectives which enriched the data collection, analysis and interpretation across diverse contexts. This is an under-researched area, and one where it is critical that evidence is generated to feed into ongoing policy and practice. We included multiple settings and voices; working across four diverse settings (Africa, Middle East, Asia) which supported the broader generalisability of the findings; with many commonalities emerging in experiences and implications. We used gender analysis frameworks which supports a rigorous and transparent approach; we synthesised and structured the findings using Morgan et al.'s framework, whilst we used Steege et al.'s framework to discuss the findings.

However, there are several limitations. We conducted the study at one particular point in the evolution of the pandemic, and we have situated our analysis and discussion appropriately. The perspectives of the community and patients are missing, which limits triangulation from diverse perspectives, and further research is required here. We have developed an in-depth understanding of the gendered experiences of CTC providers in fragile settings during COVID-19 including implications for policy and practice, and the next step is to implement and test these actions.

**5. Conclusions**

CTC providers have played a key yet often under-recognised and under-supported role in the COVID-19 response in fragile settings. Efforts should also be made to reduce the gendered challenges CTC providers face through, for example, promoting the engagement of community leaders, family members/husbands and communities to help gain greater acceptance of and support for CTC providers' roles and work duties. Understanding CTC providers' experience using a gender lens is critical to developing gender equitable approaches to support their critical roles in the COVID-19 response and future crises. These approaches should focus on support from families, communities and health systems as these all shape CTC providers experiences.

**Author Contributions:** Conceptualization, J.R., K.H., R.Y., S.T., W.M., H.W., K.K.T., A.A., A.P., S.B. and O.C.; Methodology, J.R., K.H., R.Y., S.T., W.M., H.W., K.K.T., A.A., A.P., O.C. and S.B.; Validation, J.R., K.H., R.Y., S.T., W.M., H.W., K.K.T., L.K., A.A., A.P. and O.C.; Formal Analysis, J.R., K.H., R.Y., S.T., W.M., H.W., K.K.T., L.K., A.A., A.P. and O.C.; Investigation, R.Y., H.W., K.K.T., L.K. and A.A.; Writing—Original Draft Preparation, J.R. and S.T.; Writing—Review & Editing, J.R., K.H., R.Y.,

S.T., W.M., H.W., K.K.T., L.K., A.A., A.P., S.B. and O.C.; Supervision, J.R. and S.T.; Funding Acquisition, J.R., S.T., K.H., H.W. and S.B. All authors have read and agreed to the published version of the manuscript.

**Funding:** This material has been funded by UK Aid from the UK government; however the views expressed do not necessarily reflect the UK government's official policies. This research was funded by the Foreign, Commonwealth and Development Office, grant number PO 8610.

**Institutional Review Board Statement:** We received ethical approval from the Liverpool School of Tropical Medicine (20-070), the Sierra Leone Ethics and Scientific Review Committee (09/10/2020), Social and Behavioural Sciences Institutional Review Board at the American University of Beirut in Lebanon (SBS-2020-0393), the Institutional Review Board-1, Department of Health, Ministry of Health and Sports, Myanmar (2020/35) and Nepal Health Research Council (61U2020P). All subjects gave their informed consent for inclusion before they participated in the study.

**Informed Consent Statement:** Informed consent was obtained from all subjects involved in the study.

**Data Availability Statement:** All datasets are available from the corresponding author on reasonable request.

**Acknowledgments:** We thank all the research participants who shared their stories and experiences. We thank Karen Miller for support in developing figures.

**Conflicts of Interest:** The authors declare no conflict of interest. The funders had no role in the design of the study; in the collection, analyses, or interpretation of data; in the writing of the manuscript, and in the decision to publish the results.

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
