# Peer review of "The Gendered Experience of Close to Community Providers during COVID-19 Response in Fragile Settings: A Multi-Country Analysis"

_socsci, doi:10.3390/socsci11090415_

Round 1

Reviewer 1 Report

Thank you for allowing me to review this interesting pertinent paper. This is clearly a relevant and timely study of gender inequality within the universal health coverage agenda for low and middle income countries. While the study was carried out during the most disruptive period of the Covid pandemic, this does help to highlight the equality gap, its impact and drivers, as a stressor on usual practices and contexts. It is likely to have much transferability to other emergency situations which test fragile structures and systems.  The transferability of the geographical sites was justified, and this could have been emphasised further - though possibly word limitations restricted this. The point was well made, however. 

There was good use made of extant evidence, some apparent self-referencing, but may simply show the narrowness of this specific field of research. In view of the claim for tranferability, however, I wonder if evidence could also have been used from wider sources to illustrate the often universality of the issues for local health care and gendered inequality across LMICs. But a minor point. 

The use of the frameworks appeared a useful way to scaffold the analysis, the gendered elements did not stand out clearly if this is the focus of these tools. I would have liked to know more about how power is negotiated on a gendered basis, for example, and how values are defined by gender. There was some reporting, but as a reader I had to do this work myself rather than read an interpretation. This is only to give feedback as a reader who wants to know more about the main focus rather than a criticism. 

I did take issue with a couple of phrases that were used that didn't strike the right note.

In the section on Health System, the second paragraph starts by claiming 'We need to strengthen training...' It is not clear who 'we' are but it reads as 'we, the West/HICs rather than perhaps something else the authors meant. 

Also, under the Community section heading, a sentence starts with 'We often rely on CTC performers...' and goes on to mention other community and traditional leaders, but no explanation of, again, who is 'we' or explanation of what community leaders would/should be doing, and no reference to evidence to give examples. This appears to be starting a new area of discussion that is not expanded upon. While it is a relevant suggestion, it needs further explanation or maybe its a rabbit hole best left for another paper? 

Minor punctuation errors such as missing commas and capital initials but otherwise, well written and easy to read and digest. 

I hope my feedback is as helpful as I intend it to be. 

Author Response

Dear Reviewer,

Thank you for your very useful and thoughtful feedback. Please see below our responses to the points you have raised and our revised manuscript with our track changes.

Best wishes

Authors

  1. Thank you for allowing me to review this interesting pertinent paper. This is clearly a relevant and timely study of gender inequality within the universal health coverage agenda for low and middle income countries. While the study was carried out during the most disruptive period of the Covid pandemic, this does help to highlight the equality gap, its impact and drivers, as a stressor on usual practices and contexts. It is likely to have much transferability to other emergency situations which test fragile structures and systems.  The transferability of the geographical sites was justified, and this could have been emphasised further - though possibly word limitations restricted this. The point was well made, however. 

Thank you for your positive feedback on the relevance and timeliness of the study. We have added additional detail about the transferability of findings to other crises to the last paragraph of the introduction – lines 92 and 93; and the first paragraph of the discussion – lines 625-626.  

  1. There was good use made of extant evidence, some apparent self-referencing, but may simply show the narrowness of this specific field of research. In view of the claim for transferability, however, I wonder if evidence could also have been used from wider sources to illustrate the often universality of the issues for local health care and gendered inequality across LMICs. But a minor point. 

Thank you, as stated in lines 66-27 there is indeed limited literature on gender, CTC providers and COVID-19. We have added further broader references though to demonstrate the wider relevance of the issues for health systems, in the introduction and discussion.

  1. The use of the frameworks appeared a useful way to scaffold the analysis, the gendered elements did not stand out clearly if this is the focus of these tools. I would have liked to know more about how power is negotiated on a gendered basis, for example, and how values are defined by gender. There was some reporting, but as a reader I had to do this work myself rather than read an interpretation. This is only to give feedback as a reader who wants to know more about the main focus rather than a criticism. 

Thank you for this thoughtful feedback. We have tried to make the focus on gender and values more explicit with some additional links/ signposting in the results on how gendered norms and values shape experience. 

  1. I did take issue with a couple of phrases that were used that didn't strike the right note.

In the section on Health System, the second paragraph starts by claiming 'We need to strengthen training...' It is not clear who 'we' are but it reads as 'we, the West/HICs rather than perhaps something else the authors meant. 

Thank you for picking up on this important point. It was not our intention to suggest that it is the West or HICs or the authors who should strengthen training etc. We have edited this text so that it reads: “Managers of CTC providers and those responsible for developing policies and guidelines related to CTC providers can strengthen training, access to information and supervision so that CTC providers have up-to-date, accurate and culturally appropriate information.” See lines 711-714.

Also, under the Community section heading, a sentence starts with 'We often rely on CTC performers...' and goes on to mention other community and traditional leaders, but no explanation of, again, who is 'we' or explanation of what community leaders would/should be doing, and no reference to evidence to give examples. This appears to be starting a new area of discussion that is not expanded upon. While it is a relevant suggestion, it needs further explanation or maybe its a rabbit hole best left for another paper? 

Thank you. We have added some more text here to explain that “Health systems actors such as policy makers and managers often rely on CTC providers’ individual relationships with community members to enable them to carry out their health activities” – line 674-676. We have also outlined some examples of what community leaders could do to support CTC providers – lines 678-683.  

  1. Minor punctuation errors such as missing commas and capital initials but otherwise, well written and easy to read and digest. 

Thank you. We have addressed the punctuation errors.

  1. I hope my feedback is as helpful as I intend it to be. 

Thank you for your very useful feedback, which we have addressed and believe this has strengthened the paper.

Reviewer 2 Report

This study presents information on close to community (CTC) providers during the COVID-19 pandemic. The author(s) include data from across four countries and are interested in the gendered experiences of the CTC providers. The data include a document review, in-depth interviews or focus group discussions, and key informant interviews with local stakeholders. 

The authors note there is limited training and supervision across all country contexts. Is this the first instance where CTCs were organized to work together on a medical emergency/pandemic? If so, this might explain the lack of training? Or is this something that is regularly done? A “normal” way to use CTCs? Is COVID-19 different than other sorts of medical interventions? 

There are limited equipment and supplies across countries. The authors note CTCs in Myanmar were not prioritized for vaccination. What about other countries? Seems like vaccination is different than PPE and other resources. How is this different across countries? Affected by the country context—access to vaccines? What about PPE — was this available widely in the countries? That is, were other medical providers given access to PPE while CTCs did not have access? 

In terms of resources, more context might be helpful. Are resources generally available to CTCs? That is, the author(s) note mental health services were needed but largely unavailable. Were these sort of resources available before COVID-19? Are these resources available to other medical personnel? What sorts of resources do the countries have more broadly? How might this impact the resources available to the CTCs? 

Lack of cohesiveness throughout paper/findings. For example, the family and household dynamics section — this seems more focused on the individual level, or relationships within the family. Does country context affect family/household support? How/why? This is unclear. 

It seems that some themes are similar across countries while there is more variation in others. Why might this be? The impact of country context might be explored. For example, the experience of CTCs and communities with Ebola seems like it would be important for COVID-19 experiences as well. 

Social structure, infrastructure seems like an important component in how much community support the CTCs received in each county. How does infrastructure differ? Does it matter?

Decision making practices — did this differ during COVID-19 compared to pre-COVID? How/why? Did the pandemic change the structure of how decisions were made? 

Author Response

Dear Reviewer,

Thank you for your very useful feedback, which we have addressed and believe that this has strengthened the paper. Please see below our responses to the points you have raised and our revised manuscript with track changes.

Best wishes

Authors

This study presents information on close to community (CTC) providers during the COVID-19 pandemic. The author(s) include data from across four countries and are interested in the gendered experiences of the CTC providers. The data include a document review, in-depth interviews or focus group discussions, and key informant interviews with local stakeholders. 

  1. The authors note there is limited training and supervision across all country contexts. Is this the first instance where CTCs were organized to work together on a medical emergency/pandemic? If so, this might explain the lack of training? Or is this something that is regularly done? A “normal” way to use CTCs? Is COVID-19 different than other sorts of medical interventions? 

Thank you for this useful comment. In all settings, prior to COVID-19 CTC providers would usually receive some training and supervision related to their work, although the frequency and quality of this varies across the different contexts. CTC providers identified a need for more training and supervision about COVID-19 as this was a new illness for  them, that created much fear amongst health workers as well as within their communities. We have added some further text to the discussion to make this clearer – lines 690-697.

  1. There are limited equipment and supplies across countries. The authors note CTCs in Myanmar were not prioritized for vaccination. What about other countries? Seems like vaccination is different than PPE and other resources. How is this different across countries? Affected by the country context—access to vaccines? What about PPE — was this available widely in the countries? That is, were other medical providers given access to PPE while CTCs did not have access? 

Vaccinations: CTC providers in Myanmar spontaneously raised the issue of not being prioritised for vaccines. CTC providers in other settings did not raise this issue perhaps because the COVID-19 vaccination roll out had not started in Lebanon, Nepal or Sierra Leone at the time of data collection. We have included this in lines 251-254. 

PPE: there was a shortage of PPE in all settings and particularly in the first wave of the pandemic, and priority was given to health workers in health facilities and in Nepal, CTC providers involved in contact tracing. We have included this in lines 237-240.

  1. In terms of resources, more context might be helpful. Are resources generally available to CTCs? That is, the author(s) note mental health services were needed but largely unavailable. Were these sort of resources available before COVID-19? Are these resources available to other medical personnel? What sorts of resources do the countries have more broadly? How might this impact the resources available to the CTCs? 

Thank you for raising resource availability more broadly in these contexts. We have added some text to the discussion to explain about the limited availability of resources generally and prior to COVID-19, and how the emergency situation of COVID exacerbated these resource gaps. See lines 693-699.

  1. Lack of cohesiveness throughout paper/findings. For example, the family and household dynamics section — this seems more focused on the individual level, or relationships within the family. Does country context affect family/household support? How/why? This is unclear. 

Family support was important for CTC providers in all settings. However, whether this support was provided, varied across individual CTC providers, for several reasons  - family perceptions of value of voluntary and unpaid community work, other sources of income, and how It interfered with household work. We have added this to lines 384 – 387.  

  1. It seems that some themes are similar across countries while there is more variation in others. Why might this be? The impact of country context might be explored. For example, the experience of CTCs and communities with Ebola seems like it would be important for COVID-19 experiences as well. 

Thank you for raising this point. Most themes are similar across the different country contexts. Where there are differences, we have brought these points out more and explained these using our knowledge of the country context. See below and our responses to points 2,4,6, and 7.

 CTC providers in Sierra Leone who worked through the Ebola crisis, were able to draw upon the skills they developed during this time, talking with communities, and developing trust to work during the COVID-19 crisis. We have added this to the results section, lines 407-410.

  1. Social structure, infrastructure seems like an important component in how much community support the CTCs received in each county. How does infrastructure differ? Does it matter?

Thank you for this important point. In Myanmar, Sierra Leone and Nepal, community structures were already established and working, and these existing relationships with communities, health facilities and NGOs were drawn upon to support CTC providers’ work. In Lebanon, these community structures are less evident and CTC providers did not report strong community support.  We have added some text to lines 277- 281.

  1. Decision making practices — did this differ during COVID-19 compared to pre-COVID? How/why? Did the pandemic change the structure of how decisions were made? 

We did not specifically analyse decision making processes in COVID-19 at national level, rather our analysis focused on the role of CTC providers in decision making processes and whether there were any gendered implications – see section 3.2.4 “Who decides”.  With the exception of Lebanon, CTC providers were mainly absent in decision making practices on COVID-19, and where there were decision making positions/roles they were mainly male (see for example lines 517-525 on Nepal). As outlined in the discussion (lines 726-728) “involving CTC providers, and in particular women, in planning and decision-making processes when emergencies happen is critical for effective community-based interventions.”

Round 2

Reviewer 2 Report

The authors addressed the issues I raised in the first draft. I believe the revised version reads clearly and more clearly adds to the literature. The study is important and the cross-national nature of the research is helpful for scholars to understand the onset of the COVID-19 pandemic.